# Communicable Disease Surveillance in South Africa and LMICs: A Systematic Review of Systems, Challenges, and Integration with Environmental Health

**DOI:** 10.3390/tropicalmed10110314

**Published:** 2025-11-03

**Authors:** Ledile Francina Malebana, Maasago Mercy Sepadi, Matlou Ingrid Mokgobu

**Affiliations:** 1Environmental Health Department, Faculty of Science, Tshwane University of Technology, Building 5-1, Pretoria 0183, South Africa; sepadimm@tut.ac.za; 2Office of the Campus Rector, Tshwane University of Technology, Building 5-722, Pretoria 0183, South Africa; mokgobumi@tut.ac.za

**Keywords:** communicable diseases, surveillance, disease surveillance system, South Africa, Low-Middle Income Countries, PRISMA

## Abstract

Communicable disease surveillance systems are crucial for global health security, particularly in low- and middle-income countries (LMICs) where infectious disease burdens remain high. Despite disease surveillance systems being in place, the evidence on their implementation, challenges, and integration with environmental health remains fragmented. This systematic review assesses the design, implementation, and challenges of these systems across LMICs, with a focus on South Africa and the broader Sub-Saharan African region. Using PRISMA guidelines and the PICOS framework, searches across four databases identified 325 articles published between 2010 and 2025, of which 56 (17%) were included for analysis. Thematic synthesis revealed key trends, disease priorities, and surveillance tools. South Africa contributed the highest number of articles (25%), while Sub-Saharan Africa accounted for 54% overall. COVID-19 was the most frequently studied disease (20%), followed by cholera, typhoid, and measles. The Integrated Disease Surveillance and Response (IDSR) framework appeared in 25% of articles, while District Health Information Systems 2 (DHIS2) was referenced in 11%, reflecting modest adoption of digital platforms. Reported challenges included underreporting, inconsistent case definitions, limited digital infrastructure, and weak feedback mechanisms. Although integration of environmental health was widely recommended, it was marginally implemented. Overall, LMICs surveillance systems remain constrained by operational and structural limitations, underscoring the need for digital investment, environmental indicators integration, and community-based approaches to strengthen epidemic preparedness.

## 1. Introduction

Communicable diseases such as cholera, malaria, tuberculosis, diarrhoeal diseases, human immunodeficiency virus/acquired immunodeficiency syndrome (HIV/AIDS), haemorrhagic fevers, respiratory infections such as severe acute respiratory syndrome (SARS), pandemic influenza, and coronavirus disease-19 (COVID-19) remain major public health challenges globally [1]. These diseases spread rapidly, cause high morbidity and mortality, and often lead to catastrophic social and economic impacts. Effective disease surveillance is therefore a cornerstone of outbreak prevention and control, enabling early cases detection, timely reporting, and the evaluation of interventions. To strengthen global surveillance, the World Health Organization (WHO) introduced the Integrated Disease Surveillance and Response (IDSR) strategy, particularly aimed at resource-limited settings, to guide member states in detecting, reporting, and responding to communicable diseases [2,3].

Despite these frameworks, low- and middle-income countries (LMICs) continue to face systemic challenges that hinder robust disease surveillance. These include under-resourced health infrastructures, fragmented reporting systems, inadequate laboratory capacity, and limited workforce training [4,5,6]. The COVID-19 pandemic, Ebola outbreaks in Sudan, cholera outbreaks in Haiti, Malawi, and South Africa, and the recent Mpox outbreaks across several countries have exposed the fragility of global and regional surveillance capacity [5,6]. Although LMICs are disproportionately affected, the evidence on implementation, integration of environmental health surveillance, and adoption of digital tools such as the District Health Information System 2 (DHIS2) remains fragmented [7,8]. South Africa presents a particularly relevant case: while it has a relatively robust legislative and regulatory framework, including the National Health Act and Regulations Relating to the Surveillance and Control of Notifiable Medical Conditions, it still struggles with local-level implementation gaps, underreporting, and fragmented municipal capacity [4]. This duality makes South Africa a valuable reference point for comparison with other LMICs.

This review, therefore, aims to compare and contrast the application of communicable disease surveillance systems in South Africa and other LMICs, highlighting their design, functionality, and limitations, as well as the extent of integration with environmental health. By synthesizing evidence across 56 articles published between January 2010 and March 2025, this study identifies common operational challenges, documents progress in surveillance frameworks such as IDSR and DHIS2 and draws lessons for strengthening surveillance systems in resource-limited contexts. Ultimately, the findings align with the United Nations Sustainable Development Goals (SDG) 3, Target 3.3, which commits all member states to ending the epidemics of major communicable diseases by 2030 and provides context-specific recommendations for improving epidemic preparedness in LMICs [6,9,10].

## 2. Materials and Methods

### 2.1. Study Design and Approach

This study employed a comparative systematic review methodology to identify, analyze, and synthesize evidence on the application of communicable disease surveillance and prevention systems in South Africa as well as in low- and middle-income countries. This approach provided a rigorous and transparent process for examining the application of communicable disease surveillance systems. At the same time, the comparative element enabled cross-national analysis of policy frameworks, implementation practices, and contextual challenges [11]. It was regarded as suitable for understanding patterns, similarities, and differences across countries while maintaining the rigor and transparency of systematic review principles.

Although this review is classified and conducted as a systematic review in accordance with PRISMA 2020 guidelines and the PICOS framework, it also incorporates certain characteristics typical of a scoping review. These include a broader research question, a wide range of study designs (e.g., cross-sectional, qualitative, and case studies), and an emphasis on mapping key themes, frameworks, and implementation challenges across various low- and middle-income countries (LMICs). This methodological choice was intentional, as it allowed for both a rigorous synthesis of existing evidence and a comprehensive comparative exploration of surveillance strategies. Therefore, this review is considered a comparative systematic review with scoping elements, combining the methodological rigor of systematic approaches with the breadth and contextual mapping often associated with scoping reviews [11].

### 2.2. Search Strategy

The relevant literature was sourced from several commonly used databases, including PubMed, Scopus, Google Scholar, and Web of Science, for the extensive systematic review. The search was limited to articles published in the previous 15 years, from 2010 to March 2025, to find the most relevant and recent research articles. Key search terms included “communicable diseases,” “disease surveillance in South Africa,” “disease surveillance,” “disease surveillance system,” “surveillance challenges,” and “surveillance in LMICs,” together with the Boolean operators to enhance the search [12]. Table 1 below lists the specified keywords, categorized by Boolean operators as general, specific, regional, and combined keywords, to consider all pieces of literature that cover the mentioned concepts.

### 2.3. Inclusion and Exclusion Criteria

Eligibility criteria were established based on the PICOS framework. The population “P” included national or local health systems in South Africa and other LMICs, while the intervention “I” focused on communicable disease surveillance systems and strategies. The comparator “C” involved cross-country comparisons between South Africa and selected LMICs, and the outcome “O” covered frameworks, policy adoption, integration, and challenges or limitations in implementation [12]. Finally, the study types included peer-reviewed articles published from January 2010 to March 2025. Consequently, the inclusion criteria as provided by the mentioned PICOS framework consisted of articles that focused on disease surveillance in South Africa and LMICs. In contrast, the exclusion criteria encompassed the opposite.

Inclusion criteria:Peer-reviewed articles published from January 2010 to March 2025.Articles about communicable disease surveillanceThe review focused on open-access articles onlyLow to middle-income countries, especially African countries.Full-text articles in the English language

Exclusion criteria:Articles focusing on non-communicable diseases.Non-peer-reviewed articles and peer-reviewed articles that were published before 2010.The articles with limited access or not open-accessArticles without full-text and written in any language other than English.

The rationale for country selection was based on the indicated criteria, and South Africa was included as the focal country due to its dual role as both an LMIC and a regional leader in disease surveillance policy. Its systems and its integration with the regional institutions provide a valuable case for comparison.

Furthermore, Table 2 below summarizes the elements considered for article eligibility, with a focus on the phenomenon of interest, study designs, context, and population.

Finally, the search results were imported from reference manager (EndNote) to a web-based tool (Rayyan) to assist in screening of the articles based on the eligibility criteria. The duplicate entries were removed. Two reviewers independently screened titles and abstracts. Full-text screening was then conducted using the eligibility criteria. Discrepancies were resolved by discussion and consultation with the third reviewer.

### 2.4. Data Extraction

Relevant articles were selected based on the inclusion criteria, and data were extracted using a systematic method. Essential information, including intervention type, target disease, outcomes, and effectiveness, was recorded on a standardized data extraction sheet. The retrieved data were categorized into subject areas, including communicable diseases and surveillance strategies. A standardized form was created as a Word document to extract relevant data from various databases, including Scopus, Web of Science, PubMed, and Google Scholar. Appendix A lists all the articles extracted from the mentioned sources, capturing study details, publication year, study design, study population, findings, challenges, and provided recommendations. Figure 1, under the Section 3, illustrates the selection process used to identify relevant literature. Two authors independently extracted the data to minimize errors [11].

### 2.5. Data Analysis

For analysis, qualitative synthesis was utilized to identify patterns, trends, and recurring themes in the literature. A thematic synthesis was performed to uncover recurring patterns across the included articles. Themes were developed through a deductive coding approach, guided by predefined categories drawn from the Integrated Disease Surveillance and Response (IDSR) framework, International Health Regulations (IHR) core capacities, and existing literature on surveillance system evaluations. These predefined domains included surveillance systems and strategies, disease focus, environmental integration, and operational challenges [13]. The lead reviewer manually coded the extracted content using a structured matrix, and themes were refined through iterative comparison across articles.

Quantitative data, such as percentages of disease incidence reduction and policy effectiveness, were summarized using descriptive statistics where applicable. A comparative analysis was conducted to highlight differences in intervention effectiveness across regions and policy frameworks. Thematic coding was employed to identify common challenges and gaps in current practices.

### 2.6. Quality Assessment (Appendix A)

This review has followed the PRISMA guidelines to enhance the transparency, reproducibility, and completeness of the review process, which contributes to the improved quality of the results produced [14]. The 27-item PRISMA checklist and flow diagram provided in Figure 1 also help to avoid bias that may be introduced when using the manual method of including and excluding the reviewed articles.

The methodological quality of the included articles was assessed using AMSTAR 2, as provided by Appendix A, a PRISMA-compatible tool regarded as a suitable standardized method for systematic reviews. Two reviewers independently evaluated all 16 items of the AMSTAR checklist, including seven critical domains (2, 4, 7, 9, 11, 13, and 15), as illustrated in Appendix A, to determine confidence in the study. Disagreements were discussed, with the third author assisting in resolving, and a consensus was ultimately achieved. The confidence in the results of each review was categorized as high, moderate, low, or critically low. Risk and bias were evaluated using the PRISMA 2020 Checklist, items 11 and 18, which require authors to specify the methods used to assess the risk of bias in the included articles [15].

## 3. Results

This section presents the findings from the reviewed articles, starting with an overview of the demographics, design, and sample sizes used to address the specified study objectives. It further summarizes publication trends over time, the geographic distribution of articles, and the most investigated communicable diseases. Additionally, it highlights the surveillance systems utilized by the studied countries, along with the challenges faced in these systems across various nations. Lastly, it discusses the integration of environmental health into surveillance systems.

### 3.1. Overview of Study Selection and Characteristics

A total of 325 articles were initially identified through database searches. After removing three (3) duplicates, 58 articles were outside prescribed publication year, whereas 156 articles were non-retrievable, 35 without accessible full texts, and 11 with irrelevant background articles and 16 perceived not related to the study and that did not meet the inclusion criteria (without clear study design and area), a final sample of 56 (14%) articles was included in the review, as presented in the PRISMA flow diagram Appendix A.

#### 3.1.1. Study Designs Followed

The reviewed articles (Appendix A) employed a diverse range of methodological approaches, highlighting the interdisciplinary and context-specific nature of communicable disease surveillance research in low- and middle-income countries (LMICs). A significant portion of the articles employed cross-sectional designs, which are particularly useful for evaluating healthcare workers’ knowledge, perceptions, and adherence to disease surveillance protocols. These articles were frequently applied in national or sub-national contexts, such as in South Africa, Iraq, and Nigeria, where snapshot assessments were used to identify gaps in system performance [14,15].

Another critical design category was systematic and scoping reviews, which synthesized evidence across multiple countries or health contexts. These articles evaluated the implementation of emergency preparedness initiatives, community-based surveillance systems, and broader disease surveillance frameworks, such as IDSR and District Health Information System 2 (DHIS2), which is an open-source web-based system used to collect, manage, and analyze health data as well as tracking disease trends, service delivery, and performance management in real time. They provided meta-level insights into policy effectiveness, operational structures, and regional disparities.

Qualitative articles, including key informant interviews and focus group discussions, were also prevalent. These were particularly valuable for unpacking the institutional and socio-political challenges that shape the success or failure of disease surveillance programs. For example, qualitative research conducted in the Kurdistan Region of Iraq examined the failures in integrating COVID-19 surveillance into existing national systems [15,16,17,18].

Several comparative articles examined trends across countries or regions to highlight variations in disease burden, surveillance capacity, or policy uptake. One study, for instance, compared the implementation and outcomes of disease surveillance efforts in the ECOWAS and SADC regions, identifying both strengths and deficits in regional coordination [19].

Additionally, descriptive and observational articles were employed to document specific outbreaks or epidemiological trends. These articles contributed to understanding disease patterns and health system responses in real-time scenarios, such as the diphtheria outbreak in KwaZulu-Natal or the influenza A outbreak in a South African boarding school [14,20].

Case study designs were used to document implementation processes and technological innovations. These often involve evaluations of digital surveillance systems, such as Sierra Leone’s experience with electronic case-based surveillance using DHIS2 Tracker modules [21,22,23]. Several articles employed mixed methods designs, integrating quantitative data with qualitative insights. These approaches proved particularly effective in assessing the real-world performance of IDSR systems in countries like Malawi and Guinea, providing a more comprehensive perspective on the strengths and limitations of surveillance systems [24,25,26].

#### 3.1.2. Sample Sizes

Sample sizes among the reviewed studies varied considerably, largely influenced by the respective research designs and the breadth of their investigative scope. Subsequently, the studies that utilized the national surveillance databases or adopted systematic review approaches generally included much larger datasets. For example, one review screened more than 1400 publications, while another study analyzed 648 reports drawn from Malawi’s District Health Information System 2 (DHIS2) [27,28]. Therefore, such extensive samples enabled broader analyses of disease trends, surveillance performance indicators, and policy implementation outcomes, providing valuable insights into the overall functionality and responsiveness of national surveillance systems.

In contrast, qualitative articles often engaged smaller, targeted groups of participants. The interviewees in these articles typically ranged from 14 to 57 participants, including healthcare professionals, district surveillance officers, public health policymakers, and representatives from national public health institutions. For example, the study in the Kurdistan Region of Iraq utilized 14 key informants, while another study involving five countries conducted in-depth interviews with 57 stakeholders [29,30].

Articles employing survey methodologies gathered larger, structured responses from healthcare workers. A study conducted in Nigeria included 245 physicians in a knowledge, attitudes, and practices (KAP) survey, while another involving 270 healthcare workers in Anambra State examined challenges in data collection and notification systems. Similarly, the study assessing environmental health practitioners’ (EHPs) roles in South Africa’s Ekurhuleni Municipality collected insights from 95 respondents [31,32,33,34].

At the community level, population-based articles often included hundreds of participants. In Catalonia, a study investigating a community-based surveillance model involved 677 individuals, while an influenza outbreak investigation in a South African school evaluated responses from 308 students through a combination of retrospective cohort and cross-sectional designs [35,36].

Overall, the variation in sample sizes reflects the scope and aim of each study. While qualitative and case study approaches prioritized depth of understanding and contextual richness, survey and database-driven designs aimed at breadth and generalizability. This diversity in sample composition ensured that the review captured both systemic patterns and grassroots realities in communicable disease surveillance across LMICs. This variation in study scale and sample size underscores the methodological diversity observed in CDS research. It also reflects the breadth of engagement, from localized articles to nationally representative surveillance efforts. These research activities have evolved, as shown by the publication trends in the next section.

### 3.2. Publication Trends and Geographic Distribution

#### 3.2.1. Annual Distribution of Publications

A total of 56 articles were included in this review, spanning from 2010 to 2025. Early in the period (2010–2012), publications were limited, primarily focusing on outbreak-specific investigations, such as cholera preparedness and rodent-borne diseases. However, the overwhelming majority (48 out of 56, or 86%) were published after 2013, reflecting the increasing global attention to strengthening communicable disease surveillance systems in low- and middle-income countries (LMICs). Notably, 24 articles (43%) were published between 2019 and 2022, suggesting a surge in academic output following the COVID-19 pandemic and renewed interest in public health system resilience (Figure 1). However, 2015 alone had six (11%) publications during the Ebola outbreak affecting the West African countries.

As a result, this indicates that most research emerged from 2013 onwards, with a notable increase in publications between 2019 and 2025. Additionally, Figure 1 illustrates the temporal distribution of publications, demonstrating a progressive rise in research output over the past decade. This trend highlights a growing scholarly response to epidemic preparedness and the modernization of disease reporting platforms in LMIC contexts. The ever-increasing volume of literature reflects heightened scholarly interest in strengthening disease surveillance across diverse geographic contexts.

Therefore, the following Figure 1 presents the annual distribution, growth, and trends of the reviewed literature as discussed above.

Considering the fundamental descriptive analysis of the given illustration, the Mean (Average) of the publications per year is ≈3.0, and the Standard Deviation is ≈1.73. Therefore, the SD of 1.73 indicates that the annual outputs typically varied by ±2 publications around the mean, suggesting a moderate fluctuation. The variance of ≈approximately 3.0 further confirms that while trends are visible, yearly output has not been uniform, as it has shown some notable peaks that occurred around 2015–2016 and 2019–2022, aligning with global outbreak events such as the Ebola and COVID-19 outbreaks. In contrast, the low-output years, from 2010 to 2012, demonstrated the degree of fluctuation in yearly publications. Lastly, the highest peak in 2024 reflects post-COVID evaluations and a possible shift towards digital health and an integrated surveillance platform, driving academic attention. Despite year-to-year fluctuations, the trendline confirms an overall upward trajectory, reinforcing that interest and research output in communicable disease surveillance are gradually increasing. However, the quadratic curve below (Figure 2) in green more accurately captures the actual pattern, highlighting a rise from 2019 to 2020, followed by a decline towards 2025. This reflects the COVID-19 peak in publications, with attention tapering off afterwards. The linear model oversimplifies by assuming steady growth, while the quadratic highlights the surge and decline cycle. Therefore, this suggests that CDS publications are event-driven (outbreak, crises) rather than following a smooth linear growth.

The linear fit yields a slope of ≈ approximately 0.12, indicating a slight upward trend that explains only 12% (R^2^ ≈ 0.12) of the variation. In comparison, the quadratic fit (R^2^ ≈ 0.39) explains 39% of the variation, which is significantly stronger than the linear fit.

Above all, by 2025, the cumulative count shows substantial evidence base, further reflecting increased academic and policy attention to communicable disease surveillance. Figure 3 below further confirms the surges that occurred around the years and during the COVID-19 era.

The above boxplot visualizes the spread and outliers in annual publication counts, where the line inside the box shows the median of three publications per year. Most years fall between two and four publications, with whiskers extending from low-output years of one publication to high-output years of five publications. Although the peak years (2019–2022) lean towards the upper end of the distribution, no outliers were found.

#### 3.2.2. Geographical Distribution of Reviewed Literature

This section explores the geographical scope of this research by categorizing the articles according to broader world regions made up of countries. Figure 3 shows the distribution of the articles by region, providing a combined view that complements the county-level analysis. The reviewed articles were conducted across 27 countries, with most originating from Sub-Saharan Africa. South Africa had the largest share of the reviewed literature, contributing 14 out of 56 articles (25%). This was followed by countries such as Nigeria and Malawi, each contributing multiple entries. Overall, articles from Sub-Saharan Africa made up 54% (30/56) of the sample, highlighting the region’s high burden of communicable diseases and its importance as a focus for research and investment in the global surveillance system. Regional and multi-country articles also played a significant role, especially those comparing disease surveillance implementation across economic communities or linguistic groups (e.g., SADC vs. ECOWAS). As a result, regional groupings like Southern Africa, West Africa, and East/Central Africa within Sub-Saharan Africa reinforce the strong research focus on areas with a disproportionate burden of communicable diseases (Figure 4).

Figure 5 below illustrates that Africa dominates the dataset, with 54% of articles, including regions such as Sub-Saharan Africa, Southern Africa (SADC), and West Africa (ECOWAS), as well as countries like South Africa, Nigeria, Uganda, and Guinea. The Middle East follows Africa with a representation that includes Saudi Arabia, Israel, Jordan, Iraq (Kurdistan), the Palestinian Territories, and Yemen. Asia is represented by India, Thailand, and the Philippines. Latin America includes Ecuador and Haiti. Europe is represented by only one country, specifically Catalonia, which is part of Spain. Lastly, Global/Multiregional articles account for one entry (3%).

There was also substantial representation of articles focused on broader regions such as Sub-Saharan Africa, the Middle East, and global comparisons among LMICs. The findings highlight that, despite global attention, research on surveillance systems is unevenly distributed, with more articles concentrated in a few countries that have better access to research infrastructure. While geographic representation varied, the thematic focus of articles also differed based on regional disease burden and public health priorities.

### 3.3. Communicable Diseases Focus and Surveillance System

This section explores the predominant communicable diseases addressed in the reviewed literature and the corresponding surveillance systems employed. Emphasis is placed on identifying disease-specific priorities within the national surveillance frameworks and assessing the tools and functionality of the systems used for detection, reporting, and response. The analysis provides insight into how LMICs settings align their surveillance efforts with epidemiological burdens and public health capacities.

#### 3.3.1. Most Frequently Studied Diseases

The following section discusses the most commonly addressed communicable diseases in the reviewed literature. The review revealed a wide range of diseases monitored through surveillance systems. The diseases most frequently covered in the reviewed articles were highly diverse, including acute respiratory infections, foodborne illnesses, and vaccine-preventable diseases (Figure 6). The most commonly addressed diseases were COVID-19, Cholera, Measles, Influenza A, Ebola, Listeriosis, Enteric fever (typhoid), HIV/AIDS, Tuberculosis, Candida auris, Diarrheal diseases, Anthrax, Dengue, Yellow fever, and Rabies [37,38,39,40,41,42]. COVID-19 was the most frequently studied, appearing in 7 of 56 articles (13%). Other commonly featured diseases included cholera, typhoid, measles, influenza, tuberculosis, and HIV/AIDS, each of which appeared in 2 to 4 articles (6–11%). Several articles also addressed general syndromic surveillance, foodborne outbreaks, or integrated responses without focusing on a single pathogen. The prevalence of high-impact, outbreak-prone diseases in the reviewed literature underscores the ongoing emphasis on epidemic preparedness in the public health agendas of LMICs [42]. Figure 6 presents the distribution of disease focus among 33 out of 56 included articles. Twenty-three articles focused on general surveillance architecture or community-based surveillance strategies without a specific disease.

#### 3.3.2. Surveillance Frameworks and Tools Used

Across the selected LMICs, governance structures for communicable disease surveillance varied significantly, as also highlighted by the content of Table 3. South Africa operates under a decentralized local health system with formal integration into national and regional structures. In contrast, countries like India and Bangladesh rely more on a centralized command structure, with limited autonomy at the district level. These differences affected policy responsiveness and coordination across sectors. Countries with established national public health institutes (e.g., Nigeria, Kenya) showed stronger surveillance governance and clearer reporting lines [40,41,42].

Understanding the frameworks these articles employed is crucial for assessing the infrastructure supporting disease tracking, which is explored in this section. The Integrated Disease Surveillance and Response (IDSR) strategy was by far the most frequently implemented surveillance framework across the reviewed articles [43,44,45,46,47,48]. IDSR was reported in 14 of 56 articles (25%), reflecting its widespread institutionalization in African health systems under WHO guidance, followed by the Notifiable disease system with 13% (7/56). In contrast, only six articles (11%) reported the use of DHIS2 and laboratory-based surveillance, indicating limited but growing adoption of digital surveillance tools [49,50,51,52]. Moderate frequency demonstrated on Community-based surveillance, 9% (5/56), Environmental surveillance, as well as mobile/digital tool, with 5% (3/56) each. The lower uptake of DHIS2 may reflect persistent infrastructure challenges, including unreliable internet access, lack of digital literacy, and the cost of maintaining electronic systems in resource-constrained settings (Figure 7). Several articles also referenced vertical disease-specific surveillance structures operating in parallel with national systems, often without full integration (i.e., event-based and case-based surveillance, sentinel, genomic surveillance, etc.).

The dominance of IDSR may reflect its endorsement by the World Health Organization for standardizing disease reporting in resource-limited settings. Meanwhile, DHIS2’s digital capability is still limited to better-resourced environments or pilot programs [52]. Despite the adoption of recognized systems, such as IDSR and DHIS2, the consistent implementation and effectiveness of these systems remain problematic. Figure 7 illustrates the bar graph comparing the usage of the surveillance framework across the reviewed articles.

This comparative review reveals that while many LMICs, including South Africa, have adopted frameworks such as the IDSR or equivalent systems, the implementation and integration vary widely. For instance, Table 3 below further indicates that Kenya and Brazil show strong use of real-time digital tools, while South Africa and India are still transitioning. Countries like Nigeria and Kenya leverage community-based surveillance more effectively than South Africa, where surveillance is largely led by environmental health professionals [53].

#### 3.3.3. Governance and Policy Frameworks

In many LMICs, the governance and policy frameworks guiding communicable disease surveillance are anchored in regional strategies such as the World Health Organization’s IDSR framework and the International Health Regulations (IHR 2005). These frameworks provide a standardized approach to disease detection, notification, and response, particularly in countries across Sub-Saharan Africa, Southern Asia, and parts of Latin America. However, implementation varies widely due to disparities in political commitments, intersectoral coordination, legal mandates, and resource allocation. In comparison, South Africa exhibits a relatively more structured and legislatively grounded approach to the surveillance of communicable diseases. The country operates under the National Health Act and the Regulations Relating to the Surveillance and the Control of Notifiable Medical Conditions (NMCs), which delineate responsibilities for various government levels and establish legal obligations for reporting. Additionally, South Africa integrates digital platforms, such as DHIS2, to support the management of surveillance data. Table 4 below compares the governance and policy frameworks for communicable disease surveillance in South Africa versus those of the other LMICs.

### 3.4. Comparative Analysis of the Application Process in South Africa vs. Other LMICs

However, South Africa’s communicable disease surveillance systems demonstrated a high degree of integration with the IHR (2005) and the Notifiable Medical Conditions (NMC) framework, as indicated by Table 5 below. The digital tools, such as the Notifiable Medical Conditions Surveillance System and the use of DHIS2 platforms, enhanced real-time reporting. Nevertheless, challenges persist in rural health data quality, intersectoral data sharing, and workforce shortages [54]. Subsequently, countries such as Nigeria, Kenya, Brazil, and India presented varied models. While some effectively implemented the IDSR framework, others faced fragmentation, poor infrastructure, and coordination issues. Community-based surveillance, mobile reporting tools, and the use of local health volunteers were effective in enhancing timeliness in some contexts. However, many systems suffered from underreporting, a lack of trained staff, and limited laboratory capacity [55].

Precisely, the resource allocation and system infrastructure emerged as significant differentiators. South Africa benefits from relatively stable infrastructure, including laboratory networks and provincial reporting systems. However, disparities remain at the local (municipal) level. In comparison, countries such as India, Nigeria, and Kenya reported extensively under-resourced surveillance units, often relying on donor support for their operations, which raised concerns about sustainability. The presence of integrated disease surveillance systems (IDSR) in multiple countries enhanced coordination but required substantial logistical investment [55]. All countries reviewed faced varying degrees of human resource constraints. Workforce planning, retention, and ongoing training were recurring themes that influenced implementation capacity. Table 5 indicates the application aspects in South Africa compared to other LMICs [56].

#### 3.4.1. Reporting Processes and Data Flow

The reporting structures across countries displayed differences in frequency, quality assurance, and feedback loops, as highlighted in Table 6. South Africa has a formalized local to provincial reporting system with integration into the District Health Information System (DHIS2), while others are still transitioning to electronic systems. Real-time reporting was more feasible in countries with digital tools and trained personnel. Feedback mechanisms were weak, especially in Ghana, or inconsistent in several countries like Uganda, limiting data-driven decision-making at the local level [55,56,57].

#### 3.4.2. Digital Tools and Surveillance Technologies

Digital maturity was a major factor influencing effectiveness. Table 6 highlights that South Africa and Rwanda have adopted mobile-based systems and dashboards for case reporting and outbreak alerts. Conversely, paper-based systems still predominate in many LMICs, resulting in delayed notification and response [56]. Interoperability challenges between systems used by different stakeholders 9 e.g., veterinary, EHPs, and health sectors) were noted across the board, reinforcing the need for One Health-aligned digital systems [24,58,59].

### 3.5. Challenges or Limitations in the Implementation of Surveillance Systems Across LMICs

This section outlines common operational limitations reported across different settings, as some are included in Table 3 above. All 56 articles reported one or more systemic limitations affecting surveillance performance in terms of the capacity for early detection, reporting, and timely response to public health threats, as highlighted also in the previous section. These included underreporting of cases, a lack of standard case definitions, inadequate digital infrastructure, poor feedback mechanisms, and staff shortages at the subnational level. The most frequently cited countries facing such issues include South Africa, Malawi, Nigeria, Guinea, Sierra Leone, and Iraq. Despite the presence of structured surveillance frameworks, such as IDSR, implementation fidelity was often low, particularly at the community or primary care level. The widespread prevalence of these challenges suggests that, beyond policy adoption, attention must be given to operational support, cross-sectoral collaboration, and the use of real-time data to enhance surveillance effectiveness. Table 6 below presents a description of the categorized challenges in communicable disease surveillance systems, with a focus on the countries that are most frequently affected.

These findings underscore the need for investment in training, digital infrastructure, and standardized protocols. These challenges suggest that improving surveillance requires not only system optimization but also a broader approach that incorporates environmental health determinants. The heatmap below, Figure 8, illustrates the surveillance challenges encountered by the countries included as the most frequently affected countries by Table 6 above.

Figure 8 above highlights the diversity and frequency of the most commonly reported challenges in communicable disease surveillance systems across multiple countries. South Africa emerges as the most affected, reporting six distinct challenges, including underreporting of cases, lack of standardized case definitions, inadequate feedback mechanisms, workforce shortages, limited community-based surveillance (CBS), and weak integration of environmental health into surveillance systems. This may be because the majority of the study was conducted in South Africa. This broad spectrum of challenges underscores systematic gaps at multiple levels of the surveillance infrastructure [53,57,59]. India and Nigeria also show a high prevalence of challenges, particularly in fragmented data systems, insufficient digital infrastructure, and human resource constraints. Countries such as Uganda, Malawi, and Sierra Leone face notable limitations in utilizing real-time data and accessing digital support, suggesting persistent infrastructural and technological barriers. Therefore, this heatmap not only illustrates the uneven distribution of CDS challenges but also highlights the need for targeted interventions to strengthen system components that consistently underperform across diverse contexts.

### 3.6. Integration of Environmental Health in Surveillance Systems

This section explores how integrating environmental health into surveillance systems (Table 7) could enhance public health responses. Several articles reviewed in this paper emphasize the critical and often overlooked role of environmental health in preventing and surveilling communicable diseases [53,54]. Environmental factors such as unsafe drinking water, inadequate sanitation, poor waste management, and deteriorating air quality are consistently identified as root causes of outbreaks of waterborne and respiratory infections, including cholera, typhoid, and diarrheal diseases. These findings reinforce the understanding that environmental conditions are not merely background risk factors but active components that influence the spread and recurrence of communicable diseases in low- and middle-income countries (LMICs) [53]. Table 7 below provides the status of environmental health integration in the national surveillance systems.

This table highlights the variability in integrating environmental health into surveillance systems. Countries like South Africa and Nigeria lead in integration but still face challenges in consistency and technological coordination. Meanwhile, others either lack formal mechanisms or depend heavily on external support. The inclusion of WASH indicators and EHPS in surveillance frameworks remains a key differentiator for effective disease prevention, especially for outbreak-prone and water-related diseases. This comparative view underscores the need for standardized guidelines, capacity building, and policy alignment to achieve more comprehensive and environmentally sensitive surveillance systems across LMICs [54,55].

However, some countries and regions have taken preliminary steps to integrate environmental surveillance into broader surveillance frameworks, including wastewater sampling and air quality monitoring. These efforts provide an early warning system that can detect pathogen circulation before clinical cases are formally reported. For instance, environmental sampling was employed as a proactive tool in articles evaluating the risk of typhoid transmission in densely populated urban settings [55]. However, such integration remains limited and is generally under-resourced across the LMIC context. Figure 9 emphasizes the integration of environmental health in the national surveillance systems, focusing on the three core components, which are environmental monitoring, EH practitioner involvement, and the WASH indicator inclusion.

Figure 9 provides a comprehensive overview of how the seven countries integrated environmental health components into their public health systems across the three dimensions, such as environmental monitoring, EH practitioner involvement, and WASH indicator inclusion. Kenya exhibits the highest level of overall integration across all three assessed dimensions highlighted in Figure 6, indicating a well-rounded and comprehensive approach. On the other hand, Brazil and Uganda also showed stronger practitioner involvement, suggesting robust workforce engagement, though Uganda lagged in environmental monitoring and WASH metrics. India and Thailand reflected moderate to high integration, especially in environmental monitoring and the WASH indicator, with slightly lower scores for practitioner involvement in Thailand. However, South Africa exhibited balanced yet moderate integration across all domains, suggesting areas for improvement, particularly in strengthening the roles of EH practitioners. In contrast, Bangladesh and Uganda showed the lowest overall integration, indicating potential gaps in their EH surveillance capacity. These disparities underscore the need for targeted interventions to enhance EH integration in countries with weaker performance, thereby improving the effectiveness of communicable disease surveillance and response systems [53,55].

A key observation is that environmental health is seldom treated as a core function within national communicable disease surveillance strategies. This presents both a gap and an opportunity: the lack of structured collaboration between environmental health services and epidemiological surveillance functions restricts the system’s ability to predict and control outbreaks [58]. Multisectoral collaboration, especially with environmental health practitioners, could foster more comprehensive surveillance systems that are better equipped to identify and respond to high-risk conditions in vulnerable communities.

### 3.7. Quality Appraisal and Risk of Bias

This section summarizes the AMSTAR 2 (2017) (A Measurement Tool to Assess Systematic Reviews) quality assessment findings based on Appendix A, focusing on the responses and expectations for each of the 16 items commonly used in the tool. The most important domains achieved by the assessment include domains 1, 3, 5, 6, 7, 9, 11, 12, 13, 14, and 16, which reflect a high standard of review methodology. Domains such as comprehensive literature searching (Domain 4), duplicate study selection and data extraction (Domains 5 and 6), adequate risk of bias assessment (Domain 9), and appropriate synthesis methods (Domain 11) are crucial for minimizing errors and bias. Furthermore, addressing domains related to reporting funding sources (Domain 10), conflict of interest disclosures (Domain 15), and explanation of heterogeneity (Domain 14) enhances the trustworthiness and usability of the review findings. Although Domain 2 concerns protocol registration, the consistent achievement of the remaining domains still indicates a high level of methodological quality, which strengthens confidence in the review’s reliability and reproducibility. The review process was planned objectively, and credible tools, such as the PICO strategy and PRISMA guidelines, were used to select appropriate literature. The following Pie chart (Figure 10) shows the percentage of quality ratings based on the domains assessed in Appendix A.

According to the presented results in Figure 10, 16 domains were assessed, and 94% (n = 15) of the domains were achieved, with only 6% (1) of the non-achieved. Therefore, the evaluation assumed that no critical domains (like bias appraisal or synthesis) were entirely omitted.

## 4. Discussion

### 4.1. Summary of Key Findings

This systematic review analyzed 56 articles focusing on communicable disease surveillance (CDS) in South Africa and other low- and middle-income countries (LMICs). The findings reveal that, while the overall volume of research has grown steadily since 2013, with a notable spike from 2019 to 2024, critical gaps in implementation, capacity, and integration persist across many low- and middle-income country (LMIC) contexts. The peaks of publication observed in Figure 1 correspond to global health events that occurred during those periods, for instance, Ebola in 2014–2016, Zika in 2015, COVID-19 from 2019 to 2020, and other emerging outbreaks in LMICs. At the same time, the early years (2004–2013) exhibit a static trend; however, post-2014 marks a growing interest and sporadic increases in communicable disease surveillance publications. The highest number of articles was seen in 2024, indicating a probable surge in attention to communicable disease surveillance. However, year-to-year variability suggests that external factors, such as health crises, have a significant impact on publication volume.

The most widely adopted surveillance framework was the Integrated Disease Surveillance and Response (IDSR) strategy, used in 14 articles. This reflects its institutionalization in many African public health systems, mainly due to endorsement by the World Health Organization (WHO) as a standardized approach for resource-limited countries [4]. In contrast, only five articles utilized the District Health Information System 2 (DHIS2), suggesting that digital surveillance remains constrained by technological infrastructure and operational challenges in most LMICs [56,57].

Despite variability in design and geographical focus, articles consistently report critical weaknesses, including delayed reporting, a lack of feedback loops, insufficient training, and fragmented data systems.

Consequently, this review highlights both the commonalities and contextual divergences across LMICs in their implementation of CDSP functions. While most countries adopt a form of integrated Disease Surveillance and Response (IDSR), the level of system maturity and institutional capacity varies widely. The findings suggest that South Africa has made considerable progress in digitalizing its surveillance system and aligning with international standards. Other LMICs have leveraged grassroots solutions and mobile technologies to overcome infrastructure gaps [57,58]. However, common challenges persist, including human resource shortages, inconsistent data quality (under-reporting), and system fragmentation. These are discussed further in the subsections below.

### 4.2. Systemic and Structural Challenges Undermining Disease Surveillance Systems

Across the reviewed literature, several low-and middle-income countries reported persistent structural and operational limitations in their surveillance infrastructure. Notably, articles from Nigeria, Malawi, South Africa, Guinea, Sierra Leone, and Iraq revealed a range of systemic deficiencies that compromise the effectiveness of communicable disease surveillance [59]. A recurring challenge was the absence of standardized case definitions and unclear reporting protocols, which resulted in inconsistent data capture and confusion among healthcare workers at the point of care. Additionally, the analytical capacity at the district and facility levels was often limited, which prevented the timely interpretation and use of collected data for informed decision-making [51,60].

The widespread reliance on paper-based reporting systems, combined with a lack of adequate digital infrastructure, further exacerbated these issues. In many settings, health facilities struggled with limited internet connectivity, insufficient access to computers or mobile devices, and low digital literacy among staff. This created bottlenecks in data submission, entry, and processing, contributing to delays in identifying and responding to outbreaks. Furthermore, the absence of robust feedback mechanisms from national to subnational levels hindered the flow of critical information, undermining local surveillance efforts and eroding frontline worker engagement. These constraints collectively have profound implications for the timeliness, completeness, and accuracy of surveillance data, ultimately weakening the public health system’s capacity to detect and contain outbreaks in real-time [6,7,61].

An additional structural challenge that is identified across multiple settings is the persistence of vertical surveillance systems, which operate as isolated programs focused on single diseases such as malaria, HIV/AIDS, tuberculosis, or immunization, and are frequently not integrated into the broader IDSR framework. This siloed approach leads to duplicated efforts, fragmented reporting, and an inefficient use of limited resources. The lack of integration also obscures the overall disease burden, making it challenging to develop coordinated responses to emerging threats. Inadequate digital interoperability between sectors, combined with limited data visualization tools and restricted access to real-time dashboards, further hinders the effectiveness of these surveillance systems [8,9,62].

### 4.3. Environmental Health Surveillance: A Missing Link

A key insight from this review is the underrepresentation of environmental determinants within communicable disease surveillance strategies. While environmental factors, such as unsafe drinking water, contaminated food, poor sanitation, inadequate waste management, and air pollution, have been repeatedly identified as contributors to disease outbreaks, few articles have demonstrated the proactive integration of environmental surveillance into national systems [1,10,63].

Countries like South Africa, India, and Thailand have begun experimenting with environmental sampling, such as wastewater-based epidemiology or climate-sensitive outbreak modelling. Still, these efforts remain fragmented and pilot-based [12,64]. The inclusion of environmental health practitioners, particularly at the municipal or district level, could improve detection of early warning signs and better target high-risk areas.

Given the increasing threat of climate-sensitive diseases and urban overcrowding, the review highlights the need for multisectoral collaboration that links environmental and health data streams. This is especially relevant for addressing outbreaks of waterborne diseases, such as cholera and typhoid, as well as emerging zoonotic infections [11,65].

### 4.4. Methodological Diversity and Its Implications

The included articles featured a diverse range of research designs, including cross-sectional and qualitative assessments, as well as systematic reviews and observational articles. This diversity reflects the complex and interdisciplinary nature of disease surveillance. While many articles assessed health worker perceptions or system processes, relatively few provided robust longitudinal evaluations or effectiveness metrics for specific interventions.

This lack of longitudinal data limits our understanding of how reforms or technological upgrades have translated into improved disease outcomes or surveillance performance over time. Moreover, many articles were highly localized, often focusing on a single municipality or district, which raises concerns about the generalizability of their findings across broader national contexts [12,13,65].

For research to have a more substantial policy impact, future articles should adopt multi-region designs, incorporate quantitative performance indicators (e.g., detection lag time, completeness rates), and include cost-effectiveness analyses where possible. Partnerships with ministries of health and international organizations will be critical in scaling up such evaluations.

### 4.5. Policy and Practice Recommendations

Based on this review, several policy-relevant recommendations can be drawn to enhance the design, implementation, and impact of communicable disease surveillance systems in LMICs:Strengthen integration between surveillance platforms: National ministries should harmonize vertical programs and IDSR structures to reduce redundancy and ensure comprehensive coverage. Furthermore, develop and adopt national standards for interoperability between surveillance platforms and promote regulatory frameworks aligned with IHR (2005) and Africa CDC protocols [66].Digitize and decentralize data collection: Investing in DHIS2 and similar mobile reporting platforms can enhance timeliness, particularly in remote areas. However, infrastructure must be accompanied by training and technical support.Institutionalize environmental health data collection: Countries should expand the use of WASH indicators, air quality metrics, and climate-linked early warning systems in disease surveillance frameworks.Enhance community-level reporting: Community-based surveillance (CBS), when paired with training and supervision, has shown promise for early detection of outbreaks, and should be scaled up [14].Ensure sustainable funding and capacity-building: Surveillance is often underfunded and overlooked until emergencies occur. Governments and donors should prioritize long-term financing for training, feedback systems, and performance monitoring.These actions align with SDG 3.3, which emphasizes the elimination of major communicable diseases by 2030 through resilient and equitable health systems [15,66].

By adopting these recommendations, LMICs can strengthen their surveillance systems, support public health resilience, and align more effectively with regional disease control priorities.

### 4.6. Strengths and Limitations of the Review

A key strength of this review lies in its comprehensive scope, covering 56 articles across 27 countries, and its emphasis on both system functionality and contextual challenges. However, there are limitations. First, only English-language, peer-reviewed articles published between 2010 and 2025 were included, potentially excluding relevant grey literature or non-English publications. Second, due to heterogeneity in study design and indicators, a formal meta-analysis could not be conducted. Despite these limitations, the review provides a valuable synthesis of surveillance implementation across diverse low- and middle-income countries (LMICs) contexts (Table 8). Apart from the discoveries mentioned, several articles have highlighted the common challenges faced by most LMICs, as discussed in item 3.5. The table below summarizes these challenges along with future directions or recommendations for addressing them.

This review highlights the need for context-sensitive frameworks, sustained investment, and enhanced regional collaboration in addressing the challenges above.

## 5. Conclusions

In conclusion, integrating environmental monitoring into communicable disease surveillance frameworks can significantly improve early outbreak detection and inform context-specific interventions. Strengthening environmental health infrastructure and incorporating it into national and regional surveillance strategies should be considered a public health priority in low- and middle-income countries (LMICs). Additionally, the effective integration in underperforming context, is essential for building more resilient and responsive communicable disease surveillance frameworks.

This systematic review highlights both the diversity and cohesions of Communicable Disease Surveillance System implementation across LMICs, including the considerable variation in design, as well as the challenges revealed by the articles.

South Africa’s experience provides valuable lessons on integration and digitization, while other LMICs demonstrate advanced grassroots approaches. This review further highlights the promising innovations and practices that have improved disease detection and response in specific contexts. Addressing all the gaps identified in this review could guide policymakers, healthcare workers, and international partners in building more responsive, equitable, and sustainable surveillance systems across all low- and middle-income countries (LMICs). Furthermore, a hybrid approach that blends national-level coordination with community-based strategies and technology can enhance surveillance outcomes across the region.

## Figures and Tables

**Figure 1 tropicalmed-10-00314-f001:**
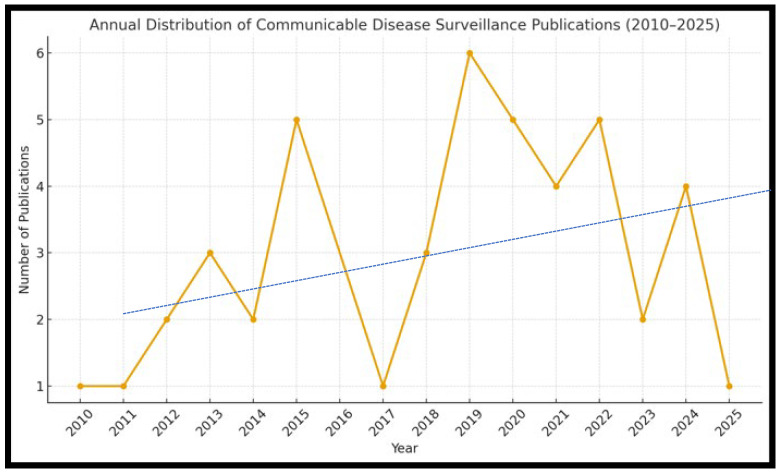
Annual Distribution, Growth, and Trends of Included Articles (2010–2025).

**Figure 2 tropicalmed-10-00314-f002:**
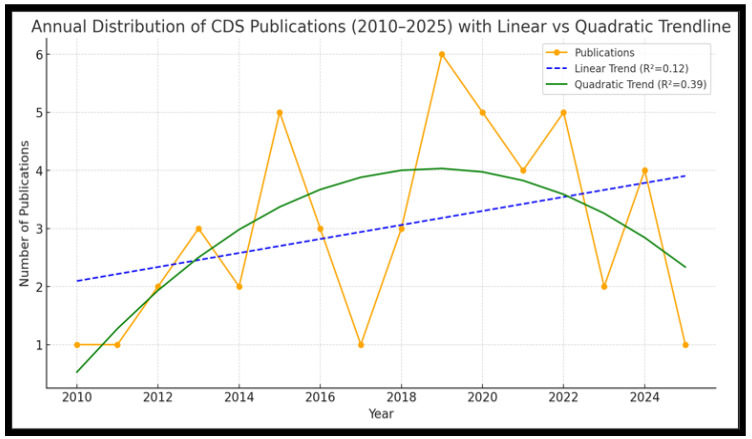
Comparison Between the Linear and Quadratic trendlines 2010 to 2025.

**Figure 3 tropicalmed-10-00314-f003:**
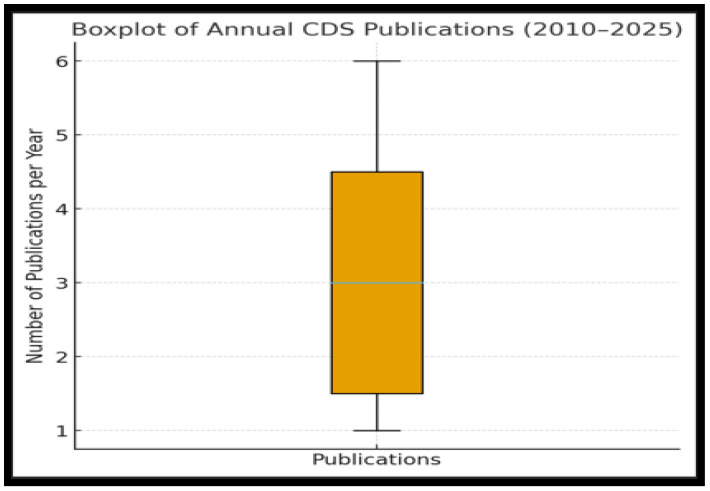
The Spread in Annual Publications from 2010 to 2015.

**Figure 4 tropicalmed-10-00314-f004:**
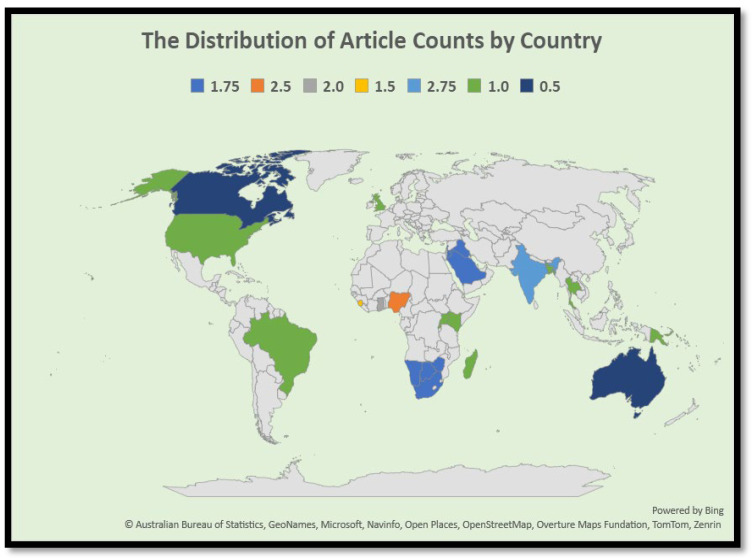
The World Map of Included Study Locations.

**Figure 5 tropicalmed-10-00314-f005:**
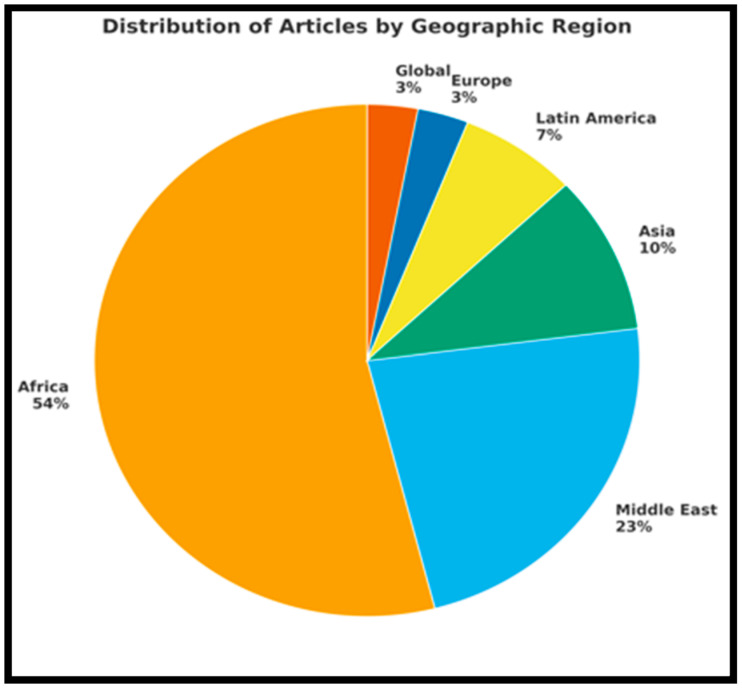
Distribution of Reviewed Articles by Geographic Region or Continent.

**Figure 6 tropicalmed-10-00314-f006:**
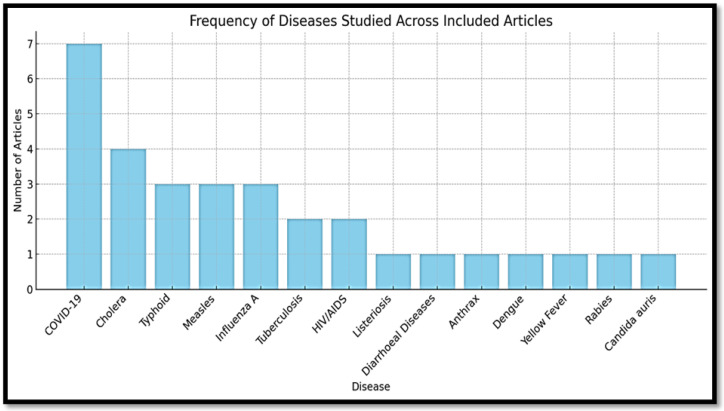
The Most Studied Diseases Across the Included Articles.

**Figure 7 tropicalmed-10-00314-f007:**
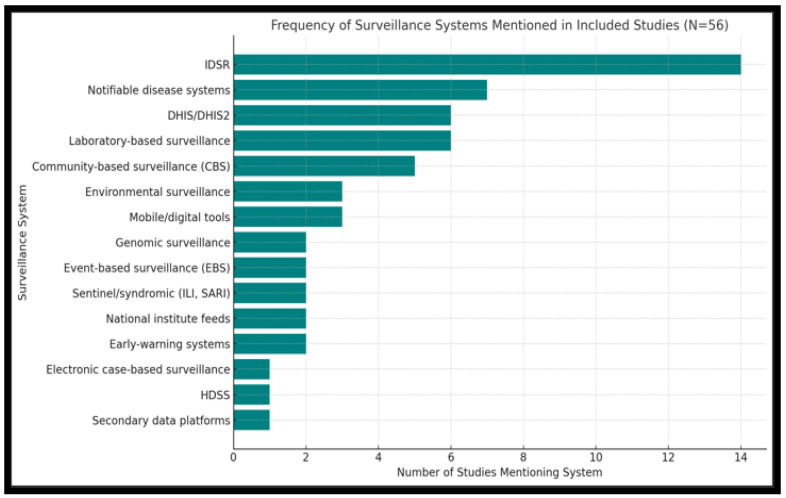
Comparative Use of Surveillance Frameworks Among Included Articles.

**Figure 8 tropicalmed-10-00314-f008:**
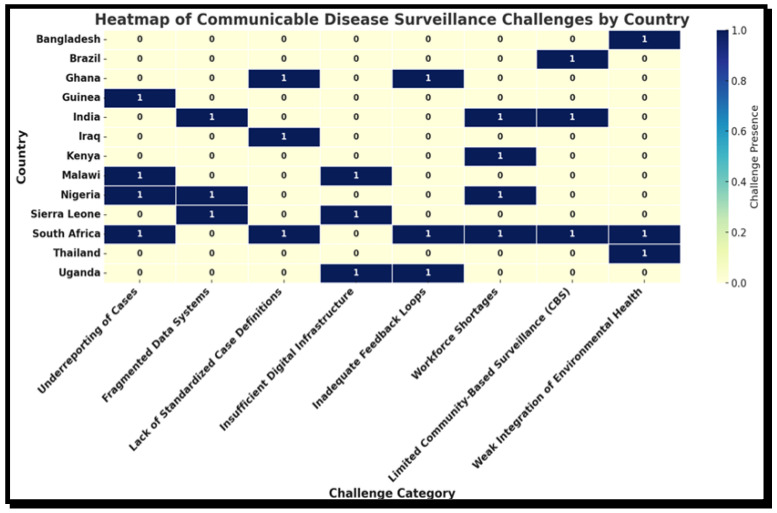
Heatmap for the Most Commonly Reported Challenges Across LMICs.

**Figure 9 tropicalmed-10-00314-f009:**
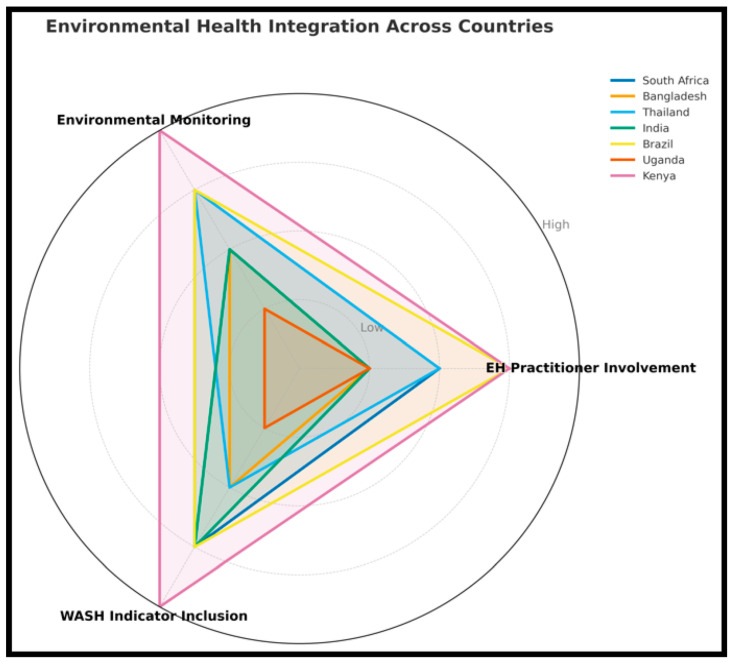
Radar Chart for Environmental Health Integration Across Selected LMICs.

**Figure 10 tropicalmed-10-00314-f010:**
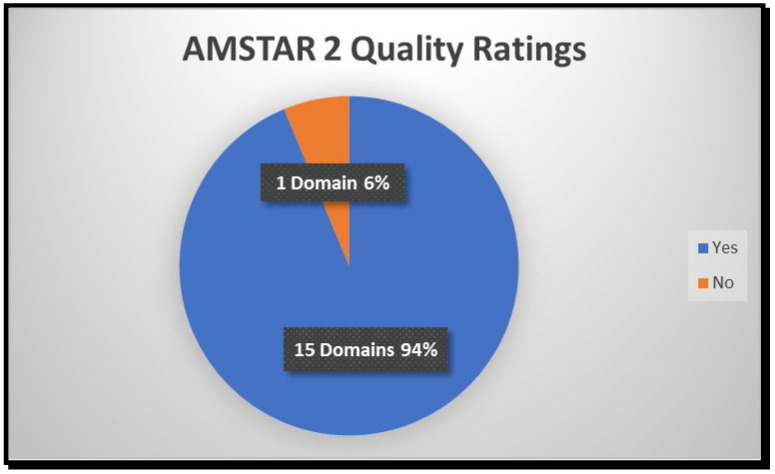
Pie Chart of AMSTAR 2 Quality Assessment Findings.

**Table 1 tropicalmed-10-00314-t001:** Search Keywords including Boolean Operators.

General Keywords	Specific Keywords	Regional Keywords	Combining Keywords
Diseases	Emerging and re-emerging communicable diseases	Emerging and re-emerging communicable diseases in South Africa	Surveillance and prevention of communicable diseases in LMICs
Disease Surveillance	Communicable disease surveillance system	Communicable disease surveillance in LMICs	LMICs’ efforts in surveillance of communicable diseases

**Table 2 tropicalmed-10-00314-t002:** Eligibility Criteria on Phenomenon of Interest, Study Type, and Design.

Criteria	Inclusion	Exclusion
Phenomenon of interest	Communicable disease surveillance systems and strategies.	Non-communicable diseases and other articles not focusing on communicable diseases
Geographical area of interest/Population	Low and middle-income countries Worldwide, including African countries and South Africa.	High-income countries, including those without such classification and not from the African continent,
Period of publication	January 2010 to March 2025.	Any other articles published before 2004 or more than 2025 years ago will be excluded.
Language	English	Articles written in languages other than English will not be considered.
Study Type	Articles	Books, and any other reports, editorial, commentaries published not as articles, and those without access to full articles.
Study Design	Review, Exploratory, guideline, case study.	Any published papers without a clear study design will not be included.

**Table 3 tropicalmed-10-00314-t003:** Comparative Analysis of Surveillance Tools and Frameworks, Real-time Use, and Limitations by Country.

Country	Surveillance Strategy/Framework	Integration with National/Regional Systems	Real-Time Data Use	Community Involvement	Key Limitations
**South Africa**	The National Notifiable Medical Conditions (NMC) system integrates IDSR elements.	Integrated with SADC and Africa CDC; aligns with IHR	Partial (improving with NMC digital platform)	Moderate—through environmental health practitioners	Fragmented municipal capacity, unequal digital access
**Nigeria**	Integrated Disease Surveillance and Response (IDSR)	Aligned with Africa CDC and WHO-AFRO IDSR guidelines	Some real-time reporting (DHIS2)	Strong community-based surveillance (CBS) is active	Inconsistent reporting, poor infrastructure
**Kenya**	IDSR + Electronic Disease Surveillance System (eDEWS)	Integrated with the East African regional response system	Strong—real-time alerts through eDEWS	Strong—community health volunteers trained	Funding gaps, training inconsistencies, and under-resourced surveillance units.
**India**	Integrated Disease Surveillance Programme (IDSP)	National-level integration, minimal international links	Moderate—uses a web-based reporting system	Limited—mostly facility-based surveillance	Overburdened system, data underreporting
**Brazil**	InfoGripe and SINAN (disease notification systems)	Integrated with PAHO; complies with IHR	Strong—especially respiratory syndromes	Moderate—CHWs play some roles in surveillance	Political instability, data transparency issues

**Table 4 tropicalmed-10-00314-t004:** Comparison of Governance and Policy Frameworks for Communicable Disease Surveillance—South Africa vs. Other LMICs.

Governance and Policy Dimension	South Africa	Other LMICs
Legal and Policy Framework	Governed by the National Health Act and Regulations Relating to Notifiable Medical Conditions (NMCs), clear legal mandate for surveillance and reporting.	Often guided by WHO frameworks (e.g., IDSR, IHR 2005), but national legislation may be weak or inconsistently enforced.
Institutional Coordination	Multi-tiered coordination across national, provincial, and local levels; integration with environmental health and municipal health services.	Variable coordination; often hampered by weak local governance and limited intersectoral collaboration.
Surveillance Strategy	Utilizes both syndromic and case-based surveillance; aligned with national priorities and regional initiatives.	Predominantly follows IDSR with adaptations; reliance on vertical programs in some contexts.
Digital Systems and Data Management	Implemented DHIS2 and electronic NMC platforms for real-time data reporting and analysis.	Increasing adoption of DHIS2 and other tools, but implementation is uneven and often lacks interoperability.
Human Resource Capacity	Stronger institutional capacity, though workforce shortages and uneven training persist.	Commonly affected by a limited public health workforce and inadequate training in surveillance functions.
Integration with International Frameworks	Aligned with IHR and Africa CDC strategies, contributes to SADC regional efforts.	Generally aligned with IDSR and IHR but limited regional collaboration in many countries.
Key Challenges	Implementation gaps at the subnational level, fragmented reporting, and resource constraints.	Governance fragmentation, underfunding, political instability, and weak enforcement of surveillance policies.

**Table 5 tropicalmed-10-00314-t005:** Comparative Analysis for Application Aspects in South Africa and Other LMICs.

Theme	South Africa	Nigeria	Kenya	India	Brazil
Governance	Decentralized	Centralized	Hybrid	National	Federal
Framework	NMC, DHIS2	IDSR	eDEWS	IDSP	SINAN
Real-time data use	Moderate	Low	High	Medium	High
Integration with EH	Limited	Moderate	Strong	Low	Moderate

**Table 6 tropicalmed-10-00314-t006:** Categorized Challenges in Communicable Disease Surveillance Systems Across LMICs.

Challenge Category	Description	Countries Most Frequently Affected	Example Citation from Review
**Underreporting of Cases**	Failure to capture and notify all relevant disease cases, particularly at subnational levels.	South Africa, Nigeria, Malawi, Guinea	[10,13]
**Fragmented Data Systems**	Parallel or vertical systems are not integrated into the national surveillance architecture.	Nigeria, India, Sierra Leone	[21,31]
**Lack of Standardized Case Definitions**	Inconsistent interpretation of diseases across health facilities.	South Africa, Iraq, Ghana	[15,48]
**Insufficient Digital Infrastructure**	Limited access to reliable internet, computers, or mobile-based tools.	Malawi, Sierra Leone, Uganda	[14,20]
**Inadequate Feedback Loops**	Poor flow of analyzed data back to local levels, weakening decision-making.	Uganda, Ghana, South Africa	[9,18]
**Workforce Shortages**	Lack of trained personnel in surveillance, particularly at the municipal or district level.	Nigeria, Kenya, India, South Africa	[30,31]
**Limited Community-Based Surveillance (CBS)**	Minimal involvement of community health workers or EHPs in early warning systems.	South Africa, India, Brazil	[29,32]
**Weak Integration of Environmental Health**	Environmental indicators not linked to communicable disease trends.	South Africa, Bangladesh, Thailand	[17,42]

**Table 7 tropicalmed-10-00314-t007:** Status of Environmental Health Integration in National Surveillance Systems.

Country	Environmental Monitoring Used	EH Practitioner Involvement	WASH Indicators Included?	Remarks
**South Africa**	Limited pilot use of wastewater surveillance (e.g., COVID-19)	Moderate—Mostly through municipal EHPs	Partially—WASH data collected but not fully linked	Some local-level integration efforts, but national systems remain fragmented
**Bangladesh**	Environmental sampling during the COVID-19 response	Low	Yes—Focus on water quality	EH indicators are included in the response, but not institutionalized long-term.
**Thailand**	Climate-sensitive modelling and environmental risk alerts	Low	No	Emerging models under public health R&D, not integrated into CDSS
**India**	Urban sanitation data used in selected districts	Low to moderate	Yes—Water, sanitation, waste	Integration is pilot-based and varies by state
**Brazil**	Zoonotic and environmental surveillance is partially integrated	Moderate—CHWs and sanitation officials involved	Partially	Some success in aligning environmental data with infectious disease alerts
**Uganda**	No structured environmental monitoring system	Low	No	EH functions are siloed from national surveillance systems
**Kenya**	Vector surveillance and waste monitoring in urban areas	Moderate	Yes	EH data is more integrated due to the One Health strategy

**Table 8 tropicalmed-10-00314-t008:** Future Research Opportunities Based on Identified Challenges.

Identified Challenges	Future Directions
Limited resources for surveillance, particularly in low-income settings.	Strengthening surveillance systems through funding or resources, and capacity building. Increasing financial and infrastructural support for disease surveillance programs.
Insufficient integration of national disease surveillance systems.	Enhancing Multisectoral Collaborations to Integrate Surveillance Efforts. Enhancing data integration across all involved sectors.
Barriers to adopting innovative tools, such as GIS, in disease prevention programs.	Adapting interventions to emerging health threats through research and innovation, and expanding research on the impact of robust surveillance on reducing disease burden.
Generally, fragmented policies exist in the surveillance and prevention of communicable diseases.	Establishing uniform policies and enforcement mechanisms. Promoting public awareness and community involvement.

## Data Availability

The original contributions presented in this study are included in the article/Appendix A. Further inquiries can be directed to the corresponding author(s).

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
