# Peer review of "Communicable Disease Surveillance in South Africa and LMICs: A Systematic Review of Systems, Challenges, and Integration with Environmental Health"

_tropicalmed, 2025, doi:10.3390/tropicalmed10110314_

Round 1
Reviewer 1 Report
Comments and Suggestions for Authors
My comments and suggestions are included in the attached report

Author Response
For A Review Article
|
Response to Reviewer 1 Comments
|
|||||
|
1. Summary |
|
|
|||
|
Thank you very much for taking the time to review this manuscript. Please find the detailed responses below and the corresponding revisions/corrections highlighted/in track changes in the re-submitted files.
|
|||||
|
2. Point-by-point response to Comments and Suggestions for Authors
|
|||||
|
Comments 1: Inclusion criteria: did the review focus on open access articles only?
|
|||||
|
Response 1: Thank you for pointing this out. We agree with this comment. Therefore, we have amended the inclusion criteria to include the fact that this review focused on open-access articles only on page 3, last paragraph, line 125, “[The review focused on open-access articles only]” and page 4, first paragraph, line 132. Therefore, the inclusion and exclusion criteria read as follows: “[Inclusion criteria: “The review focused on open-access articles only.” Exclusion criteria: “The articles with limited access or not open access]”. |
|||||
|
Comments 2: Grey literature: although it is mentioned that the article incorporated the review of grey literature, the study seemed to have only included articles from peer–reviewed journals. This could have limited the inclusion of vital information from government websites, reports and publications. |
|||||
|
Response 2: Thank you for the valuable point. We have, accordingly, revised the section on inclusion and exclusion criteria on page 3, paragraph 4, line 126, and page 4, paragraph 1, line 133 to ensure that we follow the set criteria for considering the literature as “[Inclusion criteria: “The review focused on open-access articles only.” Exclusion criteria: “The articles with limited access or not open access]”. The made changes are highlighted in red within the manuscript.
Comments 3: The article could incorporate the influence of climate change on disease outbreaks and surveillance in LMICs, an important factor that has been omitted in this article
|
|||||
|
Response 3: Thank you for the input, and we acknowledge your valuable thoughts. The authors acknowledge the importance and influence of climate change on communicable diseases. However, the study was more focused on the disease surveillance systems implemented in South Africa and in other LMICs to compare the systems’ challenges, operations, and integration with environmental health. Subsequently, Table 7 on page 18 indicates that Thailand uses the “[climate-sensitive model]” in disease surveillance. Therefore, the proposed fact will be well incorporated in the next paper that is currently in preparation.
Comments 4: Incorporate more details on political economy, governance accountability, and the long-term sustainability of funded systems to strengthen the article.
Response 4: Thank you for pointing this out. It is a valuable point. However, the proposed input might divert the focus of the study, although it is recognized for adding strength to the study.
Comments 5: Information on the publication trend analysis could be shortened to prevent the main idea of the article or public health issue on hand.
Response 5: Thank you for pointing this out. We acknowledge the input and the direction the reviewer provided however, the comprehensive publication trend analysis is important to provide the critical meta-level insights into the systems, challenges, and environmental health integration. Therefore, the shortened analysis might disadvantage the study and give less contextualization of the evidence base, research patterns, temporal shifts, and geographical gaps within the field. This analysis enhances the transparency and supports evidence mapping, consistent with PRISMA 2020 recommendations and established practices in public health.
|
|||||
|
|
|||||
|
|
|||||
|
4. Additional clarifications |
|||||
|
There is no further clarification from the authors. |
|||||

Reviewer 2 Report
Comments and Suggestions for Authors
Manuscript Title: Communicable Disease Surveillance in South Africa and LMICs: A Comparative Review of Systems, Challenges and Integration with Environmental Health.
Manuscript ID: tropicalmed-3939376
-
authors sometimeshors refer to 56 articles, and at other times to 35 or 33 articles. Please revise and correct this sentence.
-
Kindly improve the visualization of the Figure 9, Authors should be easily understand some lines are overlapping on the text, use any suitable color and improve the clarity.
-
The discussion section restates the results without analytical depth; kindly revise and correct it.
-
The paper refers to Table S1, Table S2, and Figure S1 repeatedly, but none of them are provided.
-
Many Figures 1-5, describe patterns or maps that do not appear in the documents or have no real source that needs justification.
-
This reference year does not appear to be accurate. Please revise and correct. Lebelo, K. & Van Wyk, R. (2029). Communicable Disease Surveillance in the City of Ekurhuleni: Environmental Health Practitioners’ Perspectives. 2019 Open Innovations (Oi). Ieee, 371-376. https://doi.org/10.1109/OI.2019.8908191.
-
As the year 2025 is still ongoing, how can future publications be analyzed? or fix the date
-
The paper presents the %, mean, SD, var, R2, as if derived from a meta-analysis, but these are only counts of articles; however, the use of statistical measures on publication frequencies is methodologically invalid.
-
Map references exist in the manuscript; however, they are poorly formatted. Ref (Table 4-5)
-
Please clarify the confusion between systematic and scoping reviews kindly clear it also in the manuscript.
-
Revise this section as well as “Persistent Structural Challenges in Surveillance Systems” for better clarity.
-
Kindly revise this paragraph for better clarity “Sample sizes across the reviewed articles varied significantly, shaped largely by the research design and the scope of investigation. Articles utilizing national surveillance databases or systematic reviews tended to involve large data sets. For instance, one review screened over 1,415 publications, while another study analyzed data from 648 reports drawn from Malawi’s District Health Information System (DHIS2) [27, 28]. These large samples facilitated broad assessments of trends, performance indicators, and policy implementation outcomes.”
-
Minor revisions are recommended before the final approval of the manuscript.
Author Response
For a review article
|
Response to Reviewer 2 Comments
|
||
|
1. Summary |
|
|
|
Thank you very much for taking the time to review this manuscript. Please find the detailed responses below and the corresponding revisions/corrections highlighted in red in the re-submitted files.
|
||
|
2. Point-by-point response to Comments and Suggestions for Authors |
||
|
Comments 1: authors sometimes refer to 56 articles, and at other times to 35 or 33 articles. Please revise and correct this sentence. Response 1: Thank you for pointing out this oversight. We agree with this comment and acknowledge the error made. Therefore, we have amended the article accordingly on the last paragraph of page 11, line 415, [COVID-19 was the most frequently studied, appearing in 7 of 56 articles (13%)], and page 16, paragraph 2, line 536, where 35 or 33 was changed to 56 articles with the sentence now read as follows [All 56 articles reported one or more systemic limitations affecting surveillance performance in terms of the capacity for early detection, reporting, and timely response to public health threats, as highlighted also in the previous section]. Therefore, the mentioned 35 articles was not the total number of articles considered for the study but the number of articles without full accessible text as reflected on the PRISMA flow diagram under supplementary materials, as well as on the last paragraph of page 5, line 209 of the article. The corrected contents are highlighted in red within the article. Comment 2: Kindly improve the visualization of the Figure 9, Authors should be easily understand some lines are overlapping on the text, use any suitable colour and improve the clarity. Response 2: Thank you for pointing this out. We agree with this comment. We have accordingly amended Figure 9 as found on the first paragraph of page 19, line 615 as advised, to improve visualization and clarity to the readers. Comment 3: The discussion section restates the results without analytical depth; kindly revise and correct it. Response 3: Thank you for this valuable observation. However, we acknowledge that the discussion section begins with a concise summary of the key findings of the study as per the first paragraph of page 21 from line 665, which was not intended as a repetition but rather as a structured entry point to contextualize and interpret the major findings before progressing to analytical discussion. Comment 4: The paper refers to Table S1, Table S2, and Figure S1 repeatedly, but none of them are provided. Response 4: Thank you for pointing this out. We acknowledge this comment. However, the Table S1, S2, together with Figure S1 are mentioned in the manuscript, although they are not included within the manuscript but rather forms part of the supplementary materials and indicated as such on page 5 paragraph 4, line 190-193 [Two reviewers independently evaluated all 16 items of the AMSTAR checklist, including seven critical domains (2, 4, 7, 9, 11, 13, and 15), as illustrated in Table S1 included in the supplementary material, to determine confidence in the study], the last paragraph of page 5-6, line 213 -214 [a final sample of 56 (14%) articles was included in the review, as presented in the PRISMA flow diagram (Figure S1) of the supplementary materials]. Therefore, the supplementary materials document was submitted as another attachment along with the submission of this paper. Comment 5: Many Figures 1-5, describe patterns or maps that do not appear in the documents or have no real source that needs justification. Response 5: Thank you for pointing this out. We acknowledge this observation, however the Figures 1-5 presented in the manuscript were first described in the paragraphs above the illustrations, for example, Figure 1 has the following description paragraph on the last paragraph of page 7 and the first paragraph of page 8, lines 310 -317 captured as follows “[Additionally, Figure 1 illustrates the temporal distribution of publications, demonstrating a progressive rise in research output over the past decade. This trend highlights a growing scholarly response to epidemic preparedness and the modernization of disease reporting platforms in LMIC contexts. The ever-increasing volume of literature reflects heightened scholarly interest in strengthening disease surveillance across diverse geographic contexts. Therefore, the following Figure 1 presents the annual distribution, growth, and trends of the reviewed literature as discussed above]”. Comment 6: This reference year does not appear to be accurate. Please revise and correct. Lebelo, K. & Van Wyk, R. (2029). Communicable Disease Surveillance in the City of Ekurhuleni: Environmental Health Practitioners’ Perspectives. 2019 Open Innovations (Oi). IEEE, 371-376. https://doi.org/10.1109/OI.2019.8908191. Response 6: Thank you for pointing this out. We agree with this comment. Therefore, we have amended the year accordingly as [2019] instead of 2029 appearing on the reference list of page 28, line 952 and highlighted the respective change in red. Comment 7: As the year 2025 is still ongoing, how can future publications be analyzed? or fix the date Response 7: Thank you for pointing this out. We acknowledge this comment, however the date for the reviewed articles was specifically stated in the inclusion criteria as from January 2014 to March 2025, on the last paragraph of page 3, line 124 “[• Peer-reviewed articles published from January 2010 to March 2025]” and line 103 and 119, including other sections of the manuscript on page 2, paragraph 3 line 67 were also amended to include the specific month and year of literature publication. Comment 8: The paper presents the %, mean, SD, var, R2, as if derived from a meta-analysis, but these are only counts of articles; however, the use of statistical measures on publication frequencies is methodologically invalid. Response 8: Thank you for the comment. However, the statistical measures presented in this study were not for inferential or not intended to summarize population parameters, but rather to objectively describe and visualize publication distributions. Furthermore, it is descriptive and exploratory, aimed at summarizing and quantifying the patterns, heterogenicity, comparison or relationships within the evidence base. Comment 9: Map references exist in the manuscript; however, they are poorly formatted. Ref (Table 4-5) Response 9: Thank you for pointing this out. We agree with this comment. Therefore, we have amended the format of Table 4 and 5 on the second paragraph of page 14, line 488 and the last paragraph of page 15, line 512 to meet the requirements of the article. The changes made are highlighted in red within the manuscript. Comment 10: Please clarify the confusion between systematic and scoping reviews kindly clear it also in the manuscript. Response 10: Thank you for pointing this out. We acknowledge the provided comment. Although this study followed the methodological rigor of a systematic review including adherence to PRISMA 2020 guidelines, PICO framing, and the formal quality appraisal, it also incorporated certain exploratory elements typical to a scoping review as stated on the last paragraph of page 2 lines 87-89 “[Although this review is classified and conducted as a systematic review in accordance with PRISMA 2020 guidelines and the PICOS framework, it also incorporates certain characteristics typical of a scoping review]”. With that view, this review is best described as a systematic review with scoping characteristics, and it is appropriate due to the heterogeneity, complexity, and diversity of the evidence on communicable disease surveillance in South Africa and low-and middle-income countries. Comment 11: Revise this section as well as “Persistent Structural Challenges in Surveillance Systems” for better clarity. Response 11: Agree. We have, accordingly, revised the section by refining the subtitle and amending some content. Although the paragraph is already rich in content and evidence, the changes were effected for greater clarity, conciseness, and logical flow. The subtitle is therefore changed to “[The Systemic and Structural Challenges Undermining Disease Surveillance]” on page 21, paragraph 5, line 699. Further amendments were also made to the first statement of the content on the same page, paragraph 5, line 700 to 701 stating that “[Across the reviewed literature, several low-and middle-income countries reported persistent structural and operational limitations in their surveillance infrastructure]”, and on another, second paragraph of page 22, line 720 to line 723 of this section stating that “[An additional structural challenge that is identified across multiple settings is the persistence of vertical surveillance systems, which operate as isolated programs focused on single diseases such as malaria, HIV/AIDS, tuberculosis, or immunization, and are frequently not integrated into the broader IDSR framework]”, to align with the revised subtitle. Comment 12: Kindly revise this paragraph for better clarity “Sample sizes across the reviewed articles varied significantly, shaped largely by the research design and the scope of investigation. Articles utilizing national surveillance databases or systematic reviews tended to involve large data sets. For instance, one review screened over 1,415 publications, while another study analyzed data from 648 reports drawn from Malawi’s District Health Information System (DHIS2) [27, 28]. These large samples facilitated broad assessments of trends, performance indicators, and policy implementation outcomes.” Response 12: Thank you for the comments. Agree. We have, accordingly, modified the Sample Size section to emphasize this point. The amendments were implemented to clarify, and to improve sentence flow for better comprehension of the readers as follows “[Sample sizes among the reviewed studies varied considerably, largely influenced by the respective research designs and the breadth of their investigative scope. Subsequently, the studies that utilized the national surveillance databases or adopted systematic review approaches generally included much larger datasets. For example, one review screened more than 1,400 publications, while another study analyzed 648 reports drawn from Malawi’s District Health Information System 2 (DHIS2) [27, 28]. Therefore, such extensive samples enabled broader analyses of disease trends, surveillance performance indicators, and policy implementation outcomes, providing valuable insights into the overall functionality and responsiveness of national surveillance systems]”. These amendments will be found on the last paragraph of page 6-7 of the manuscript, line 260-268. |
||
|
3. Response to Comments on the Quality of English Language |
||
|
Point 1: The quality of the English language could be improved to more clearly express the research |
||
|
Response 1: All the authors were involved in editing the comments and proofreading the manuscript to improve the quality of the English language. Furthermore, one of the language editors have assisted the authors to improve the quality of English language.
|
||
|
4. Additional clarifications |
||
|
The manuscript was proofread after amendments to ensure better logical flow, conciseness, and more clarity. |
||
